# Control and Entropy Analysis of Tip Leakage Flow for Compressor Cascade under Different Clearance Sizes with Endwall Suction

**DOI:** 10.3390/e22020128

**Published:** 2020-01-21

**Authors:** Botao Zhang, Bo Liu, Changfu Han, Hang Zhao

**Affiliations:** School of Power and Energy, Northwestern Polytechnical University, Xi’an 710129, China; zhangbo_tao@126.com (B.Z.); hanchangfu1996@163.com (C.H.); zhaohang@mail.nwpu.edu.cn (H.Z.)

**Keywords:** compressor cascade, tip clearance size, tip leakage flow, boundary layer suction, entropy analysis

## Abstract

To investigate the influence of the change of tip clearance size on the control effect of the endwall suction, the effects of endwall suction on the aerodynamic performance of the axial compressor cascade were studied numerically. Three tip clearance sizes of 0.5% h, 1.0% h, and 2.0% h (h is the blade height) were mainly considered. The results show that the endwall suction scheme whose coverage range was 8–33% axial chord can reduce the leakage flow and improve the aerodynamic performance by directly influencing the structure of tip leakage vortex. The overall total pressure loss coefficients of the three clearance size schemes at 0° angle of incidence with 0.4 inlet Mach number are reduced by about 10.3%, 10.8%, and 6.0%, respectively, at the suction flow rate of 0.7%. Under the same suction flow rate, the onset position of the tip leakage vortex of the cascade with small clearance is shifted from the 15% of the axial chord length of original to the 48% of the axial chord length, which with large clearance is nearly no changed. The leakage flow rate and the distance from the leakage vortex to the suction slot are the main reasons for the different control effect of the endwall suction under different tip clearance sizes. The difference of the spanwise distribution of flow field parameters may also cause the difference of flow control effect.

## 1. Introduction

To prevent friction and collision between the compressor rotor tip and the casing as well as between the blade root of the cantilever stator and the hub, the clearance is introduced. Some fluid flows from the pressure surface (PS) side to the suction surface (SS) side to form the tip leakage flow (TLF) driven by the pressure difference, which is affected by the main flow and the boundary layer of the endwall, and then develops downstream in the form of the tip leakage vortex (TLV). The development of TLF and TLV is shown as Figure 1. The results show that the endwall loss caused by tip clearance accounts for ~20–40% of the total loss, which has a serious impact on the overall performance of the compressor [1,2,3]. With the increase of the compressor blade load, the transverse pressure gradient increases, so does the leakage flow intensity, which has a negative impact on the compressor flow field. Thus, the leakage flow has become one of the major factors restricting the engine performance.

The influence of gap leakage on compressor is mainly reflected in two aspects, one is the leakage related loss and the other is the passage blockage near the end region. Moore’s research [4] on an axial-flow fan shows that the influence of clearance on compressor performance ranges from 70% blade span to the blade tip. With the increase of the tip clearance size (TCS), the compressor aerodynamic performance decreases and the stall margin decreases simultaneously. The results by Inoue and Lange et al. [5,6,7,8] show that the TCS is very important to the development of TLF and TLV. With the increase of TCS, the onset position of the TLV moves to the downstream, the scale and strength of TLV were increased, and the radial influence area was expanded. In addition to the above mentioned internal flow loss, the unsteady phenomena of TLF, such as the leakage vortex breaking [9,10] and the self-excited unsteady fluctuation [11,12], are closely related to the flow instability such as the rotating stall.

Because the TLF has a serious adverse effect on the compressor, it is an important and meaningful topic to control the TLF and reduce the leakage loss. The boundary layer suction (BLS) technique can improve the performance of the compressor and increase the flow capacity by opening the suction holes or slots at reasonable positions to aspirate the low-energy fluid. The BLS was first applied to control the flow separation on the blade SS in the compressor [13,14], and then it also showed a good application prospects in controlling the flow separation of hub corner [15,16,17]. In recent years, this technique has also been used to control the TLF in the clearance, and it can be divided into two categories according to different aspiration positions: one is the blade tip suction (BTS) and the other is the endwall suction (ES). Mao et al. [18,19] studied the influence of the axial position of the suction slot on the TLF of the compressor cascade controlled by the BTS and ES respectively. The results showed that both aspiration schemes could effectively reduce the TLF intensity and flow loss of the compressor cascade. In addition, the best axial position of the suction slot of the BTS scheme is downstream of the onset position of the TLV, whereas the best axial position of the suction slot of the ES scheme should include the onset position of the TLV and the surrounding area. Not only the compressor cascade is selected to research, but also the control schemes of BLS on TLF are studied in the compressor stage, which effectively inhibits the development of the TLF and TLV [20,21,22,23].

In the real work of the engine, due to the influence of various loads, temperature, and transition speed, the TCS may change accordingly. Is the suction scheme designed for the static clearance at the design point still applicable when the TCS changes? What happens to the suction effect? These are the questions that this paper focuses on and will solve. Based on the above analysis, this paper takes a high load axial diffuser cascade as the object, and mainly studies the control effect of the ES scheme on TLF under different TCS through numerical simulation, which provides a reference for the design of adsorption type compressor.

## 2. Investigated Compressor Cascade and Numerical Method

### 2.1. Compressor Cascade Description

A typical high load axial-flow compressor cascade is selected to study the effectiveness of the ES scheme under different TCS. Table 1 shows the main design parameters of the cascade, and Figure 2 shows the two-dimensional layout and some parameter descriptions of the cascade. In the current study, the TCS are given as 0.5% h, 1.0% h, and 2.0% h (that is, the TCS are given as 0.5%, 1.0% and 2.0% blade height), respectively, corresponding to the small clearance, common clearance, and large clearance.

### 2.2. Numerical Simulation Methods and Validation

The NUMECA/Fine 10.1 based on the EURANUS solver is used to solve the three-dimensional RANS equation. The central difference method and the fourth-order Runge–Kutta iterative method are used to discretize the space and time terms of the control equations. According to the previous research experience of our research group [18,19], the one-equation S-A turbulence model has the better convergence and stability than the other two equation models, and it is also found that this turbulence model can accurately predict the flow in the cascade, compared with the experimental results. Therefore, the S-A turbulence model is selected to simulate the internal turbulence in current research. At the same time, the multigrid technique and implicit residual method are used to accelerate convergence. The inlet boundary condition gives the axial intake and total parameters (101,325 Pa, 288.15 K), and the outlet of cascade passage gives the mass flow rate. For the suction scheme, the absolute suction mass flow and initial backpressure are given at the outlet of the suction slot. In addition, the adiabatic and no-slip condition is applied for the solid wall.

The NUMECA/autogrid5 10.1 is used to mesh the passage of the cascade, and a single channel O4H topology is generated. The butterfly grid technique is used in the real blade tip clearance. According to the reference [19], when the number of clearance nodes exceeds 17, the influence of the number of clearance nodes on the flow field is not obvious. In this paper, 21 grid nodes are given along the radial direction to capture the flow field within the gap. Here, the grid independence is verified to ensure the accuracy of this method. Figure 3 shows the trend of the overall total loss coefficient ($\bar{\omega }$) and inlet Mach number ($Ma_{1}$) with the total number of passage grids. It can be seen that when the total number of grids exceeds 1.06 million, especially 1.3 million, the change of grid number will not have a significant impact on the calculation results. Therefore, the total number of passage grids in the prototype cascade without suction is set to about 1.3 million in this paper.

Figure 4 shows the calculation domain setting of the compressor cascade and the calculation grid of the lower endwall and the blade SS under the TCS of 1.0% h, and the local grid enlargement at the leading and trailing edge (LE, TE) of the blade tip is shown. The inlet passage extends to 1.0 axial chord (Ca) upstream of the LE. The outlet passage reaches 1.5 Ca downstream from the TE of the blade. The grid size of the first layer near the solid wall is set to $1\times 10^{-6}m$, making the dimensionless parameter $y^{+}<2.5$, so as to meet the requirements of the turbulence model selected for numerical calculation [24].

To verify the effectiveness of the numerical calculation method used in current study, the comparison between the numerical calculation results and the experimental results without TCS is shown in Figure 5. The left and right figures show the static pressure coefficient (Cp, defined as the ratio of local static pressure to the inlet total pressure) at 50% blade span and the outlet flow angle ($\beta_{2}$) along the spanwise distribution under the conditions of 0° inflow incidence angle (i) and 0.6 inlet Mach number, respectively. Because the flow field of the cascade without clearance is symmetrical along the mid-span, the distribution of outlet flow angle is only compared in half-span. The static pressure of SS and PS of the blade is obtained by distributing the static pressure holes in the middle span of the blade in the experiment, and the outlet flow angle is obtained by the probe behind the cascade. The detailed cascade and test data can be referred to the work in [19]. According to the comparison results in the figure, although there are differences between the numerical simulation results and the experimental results, the distribution law of the two is in good agreement, and the overall error is within an acceptable range. It proves the feasibility of the numerical calculation method used in current research and the accuracy of the subsequent corresponding conclusions.

## 3. Aspiration Scheme Design

In this paper, the influence of BLS on the aerodynamic performance of the axial diffuser cascade is studied by adjusting the TCS of the cascade, and the main purpose is to explore the change of complex flow field near the tip of blade. The design of the suction scheme is based on the calculation of the original cascade flow field. The onset position of TLV of prototype cascade is 15% Ca and 28% Ca for the TCS of 0.5 h and 2.0 h at the incidence angle of 0°, and 7% Ca, and 20% Ca at the incidence angle of 4°, respectively. According to the onset position of TLV, the ES scheme in this paper is designed to control the TLF as shown in Figure 6. The suction slot is located on the upper endwall of the cascade (casing) and distributed in the range of 5 to 35% Ca of the blade. This range corresponds to the onset position of the TLV and some areas nearby. The suction slot is parallel distributed with the camber line of the blade as the centerline, and the width and height of the slot are both 1mm. The suction slot grid uses the NUMECA/IGG module to generate an H-type topology; 101, 29, and 29 grid nodes are, respectively, arranged along the axial direction, spanwise, and pitchwise, so the total number of the suction slot grids is about 85,000. The suction slot mesh is connected with the mesh of blade passage by using the full non-matching connecting technique.

## 4. Results and Discussion

First, the effect of the ES scheme on the overall performance of the cascade with three clearance sizes is discussed. In this paper, the total pressure loss coefficient (ω) is defined as Equation (1) and used to evaluate the overall performance. The overall total pressure loss coefficient is obtained by mass averaging the total pressure loss coefficient in the pitchwise and the spanwise on the cascade outlet.
(1)$$\omega =\frac{p_{t1}-p_{t2}}{p_{t1}-p_{1}},$$
where the $p_{t}$ is the total pressure; p is the static pressure; and 1 and 2 represent the inlet and outlet, respectively.

Figure 7 shows the variation of the overall total pressure loss coefficient of the cascade with suction flow rate (SFR) at the inlet Mach number of 0.4. The SFR is expressed as the ratio of the mass flow at the outlet of the suction slot to the mass flow at the inlet of the cascade passage, and the dotted lines are expressed as the loss under the same inflow condition of the prototype cascade without suction at each TCS. When the inflow incidence angle is 0°, the overall total pressure loss coefficient almost decreases linearly with the SFR, in which the slope corresponding to the TCS of 1.0% h is the largest, and the overall loss coefficient corresponding to the TCS of 2.0% h changes most smoothly with the SFR. When the inflow incidence angle is 4°, the total loss coefficient decreases with the increase of SFR, but that corresponding to the TCS of 2.0% h is likely to increase when the SFR is much larger. Different from the nearly linear correlation between the overall loss coefficient and the SFR at inflow incidence angle of 0°, the small SFR has a less influence on overall loss coefficient at inflow incidence angle of 4°. 

For quantitative explanation, Figure 8 shows the relative change of the overall loss coefficient with the TCS when the SFR is 0.0% and 0.7% at the inflow incidence angles of 0° and 4°, respectively. When the ES scheme is not carried out, the effect of the suction slot on the flow field is similar to that of the casing treatment. The overall loss coefficient is increased slightly due to the interference effect on the flow field in the end area under most cases. However, it is found that the interference effect may also reduce the loss through the investigation of the 4° inflow incidence angle under the TCS of 2.0% h. These two seemingly contradictory results can also be found from the conclusion obtained in the research about the case treatment [25]. For the cases with the SFR of 0.7%, the overall loss coefficients of the cascade with TCS of 0.5% h, 1.0% h, and 2.0% h decrease by approximately 10.3%, 10.8%, and 6.0%, respectively, at 0° incidence angle, and decrease by approximately 6.0%, 6.8%, and 3.2%, respectively, at 4° incidence angle. Combined with the above analysis, it can be preliminarily concluded that the endwall suction scheme adopted in this paper has a better effect on improving the overall performance of the cascade of which the TCS is 0.5% h and 1.0% h. However, the effect of modifying the flow field in the large clearance of 2.0% h is the worst of the three TCS schemes.

Next, the effect of TCS on tip leakage control by the ES scheme was studied with the TCS schemes of 0.5% h and 2.0% h, respectively, under the condition of inlet Mach number of 0.4 and incidence angle of 0°. Figure 9 shows the static pressure coefficient distribution on the upper endwall of the prototype and suction schemes under two clearance sizes. The black dot line starting from the minimum static pressure point in the figure is the line of the static pressure chute, which is used to represent the movement trajectory of TLV [5,26]. The onset position of the TLV of the cascade with the TCS of 0.5% h and 2.0% h is located at approximately 15% Ca and 28% Ca, respectively, that is to say, with the increase of the TCS, the onset position of the TLV moves downstream. In addition, with the increase of the TCS, the trajectory of the TLV deviates away from the blade SS as well as the strength and influence range of the TLV increase. After suction, the onset point of the TLV corresponding to the small clearance moves downstream to ~48% Ca, and the influence of TLV on the flow field is obviously weakened. The ES scheme does not change the onset position of the TLV corresponding to the large clearance, only makes its trajectory slightly close to the blade SS. Moreover, looking at the distribution of the static pressure coefficient near the cover position of the slot, the ES scheme almost affects the thickness of the whole blade under the small clearance. Nevertheless, the influence range of the suction slot under the large clearance is only limited to its own cover region, which also leads to the small influence of the ES scheme on the trajectory of the TLV under the large clearance.

Figure 10 further shows the axial velocity ($V_{z}$) contour of S1 slice at 97% span under two clearance sizes. In the figure, the area with low axial velocity indicated by dark blue circled by black ellipse dotted line is the area affected by leakage vortex. The blocking effect of leakage vortex on flow field results in the decrease of axial momentum of this part of fluid. It is basically consistent with the analysis in Figure 9 that the circumferential influence range of the low-speed region under the large clearance is larger than that under the small clearance and closer to the upstream, that is, the TLV under the large clearance has a greater influence on the flow field. The area of the low velocity region decreased obviously after the ES. In addition, the low-speed region moves downstream under the suction condition with the small clearance, whereas there is no obvious movement with the large clearance, which is also consistent with the conclusion in Figure 9.

The traditional total pressure loss coefficient used in cascade analysis does not consider the partial loss caused by the temperature change when analyzing the detailed flow field. Because the mixing effect between the fluids in the leakage flow cannot be ignored, it is inaccurate to analyze the details of leakage flow with this coefficient. Yoon et al. [27] suggested that entropy (s) is a better method to solve the problem, and in the literature [28,29], the entropy analysis method was used to analyze the flow field of cascade with severe flow separation. Then, the detailed comparative study of the flow field development under different working conditions would be carried out by using the entropy analysis method. Equation (2) shows the definition of the entropy.
(2)$$s=\frac{C_{p}}{\gamma }\log\left({\left(\frac{p}{p_{ref}}\right)}^{\left(1-\gamma \right)}{\left(\frac{T}{T_{ref}}\right)}^{\gamma }\right),$$
where $C_{p}$ is the specific pressure heat capacity and γ is the specific heat ratio.$\;p_{ref}$ and $T_{ref}$ are the reference pressure and temperature, respectively. In this paper, the total inlet parameters (101,325 Pa and 288.15 K) are selected as the reference values.

Figure 11 shows the entropy distribution of the tip region on 15 axial slices from the blade LE to 140% Ca under various conditions, and some three-dimensional leakage streamline is added. The high entropy region near the blade SS in the figure is the region affected by the TLV. In the process of downstream development, the region expands gradually, whereas the peak value of the entropy decreases. For the ES scheme compared with the prototype, the high entropy region caused by the TLV moves towards the blade SS and the area is reduced, and the entropy value is also reduced to a certain extent. It is indicated that the ES scheme can modify the tip flow field by directly weakening the strength and influence range of the TLV. The position where the area of the high entropy began to expand in the small clearance moves obviously to the blade TE after aspiration, which corresponds to the movement of the onset position of the TLV. This is consistent with the onset position of the TLV, which has no change with or without suction under the large clearance scheme; the position where the high entropy region began to expand has no obvious change, too. A much smaller region of high entropy adjacent to the TLV is caused by the passage vortex (PV), which is shown in the lower two figures as a red dotted ellipse.

To further illustrate the influence of TCS on the suction effect, Figure 12 shows the entropy and two-dimensional streamlines distribution from 70% blade span to the tip of the S3 slice near the TE of the cascade blade. This slice further confirms the influence of suction on the TLV and PV analyzed in Figure 11. In addition, it can be seen that the core of TLV is farther away from the endwall under the large clearance, and the PV is stretched into an ellipse with the core closer to the upper endwall under the small clearance after suction. Compared with the area where entropy value is more than 10 (mainly represented in green, yellow and red), the area corresponding to the TCS of 0.5% h with suction is almost reduced to half of the original, and is limited to the upper endwall corner, thus greatly reducing the blockage of the tip passage. The area corresponding to the TCS of 2.0% h is only reduced by about 20%. According to the analysis of Figure 9 and Figure 12, the difference of suction effect under the small and large clearance is mainly caused by the difference of change degree before and after suction, including the onset position and influence range of the TLV. The reason for the different degree of change is that the distance between the core of the TLV and the endwall is different. That is to say, the distance between the suction slot and the TLV increases with the large gap, and less fluid that originally located in the core of the TLV was sucked in the same SFR, which results in the weakening of the control ability of the suction slot to the TLV and the ability to modify the flow field.

To more intuitively compare the vortex structure scale and position changes under different TCS with and without suction, Figure 13 shows the TLV structure under the vortex criterion Q, and shows it with the normalized helicity. See Equation (3) for the definition of Q. The isosurface of $Q=5\times 10^{6}s^{-2}$ is selected in the figure below,
(3)$$Q=\frac{1}{2}\left(\Omega_{ij}\Omega_{ij}-S_{ij}S_{ij}\right),$$
where $\Omega_{ij}=\left(u_{ij}-u_{ji}\right)/2$ and $S_{ij}=\left(u_{ij}+u_{ji}\right)/2$ are the spin tensor and strain-rate tensor, respectively. The normalized helicity ($H_{n}$) is defined as Equation (4), it can easily grasp the direction of vortex without considering the attenuation of vorticity in the flow direction, and then conduct quantitative analysis on the properties of vortex [9].
(4)$$H_{n}=\frac{\bar{\xi }\cdot \bar{u}}{\left|\bar{\xi }\right|\cdot \left|\bar{u}\right|},$$
where $\bar{u}$ and $\bar{\xi }$ denote vectors of the flow velocity and the vorticity, respectively. The magnitude of the normalized helicity tends to unity in the vortex core, and its sign indicates the direction of swirl of the vortex relative to the streamwise velocity component.

In Figure 13, the vortex structure near the SS of the blade with the positive sign of the normalized helicity is the TLV, and the vortex structure close to it with the negative sign of the normalized helicity is the PV. The scale of TLV with large clearance is larger than that with small clearance, especially in the axial and pitchwise directions. For the small clearance scheme with suction, the structure of the TLV is attenuated rapidly, the normalized helicity is decreased, and the onset position of the TLV is moved backward to ~50% of the chord length. For the large clearance scheme, the influence of suction on the structural scale of the TLV can be ignored, but the normalized helicity value of the TLV is smaller than that of the prototype, which indicates that the intensity of the TLV is weakened, which is consistent with the above analysis. In addition, it should be noted that the separation vortex structure near the TE tip of the blade SS under large clearance is separated from the TLV structure due to the interaction of reflux, boundary layer and TLF, and the vortex structure is attenuated under the suction.

Figure 14 shows the static pressure coefficient distribution at 97% blade span under the prototype and suction conditions with two TCS. One can see that the pressure distribution on the SS of the suction slot coverage and nearby area were changed obviously by slotting along the flow direction on the upper endwall, and also has a certain impact on the pressure distribution of the blade PS, thus affecting the distribution of the blade tip load. For the condition with the TCS of 0.5% h, the static pressure coefficient of the SS from the LE of the blade to 23% Ca decreases after suction, and the blade load increases, which is caused by the local acceleration effect of the TLF due to the suction. After that, in the range of 23 to 65% Ca, the static pressure coefficient of the blade SS increases, the blade tip load decreases obviously, the blocking effect and the TLF intensity are weakened, which is the same as the conclusion observed in Figure 11. This range has also become the main action area of the suction scheme used in current study to control TLF. Besides, the decrease of the TLF intensity and the attenuation of the TLV make the static pressure coefficient of the blade PS increase in small range.

For the same inlet condition with the TCS of 2.0% h, the variation law of the static pressure coefficient distribution of this blade span caused by suction is basically the same as that of the small clearance. The static pressure of the SS of the blade decreases from the LE to 16% Ca, then increases from 16% Ca to 62% Ca, and almost does not change until the TE of blade. It is also similar to the increase of static pressure in some areas of blade PS after suction under the small clearance, the static pressure of the PS of the blade increases in the range of 16 to 65% axial chord length under the large clearance with suction. Different from the case of small clearance, the suction reduces the static pressure of the PS near the blade LE, but the blade load near the LE increases, which is the same as the suction effect on the blade tip load under the case of small clearance.

According to the work in [23], the angle between the TLF at the gap exit and the core flow can be selected to evaluate the intensity of local TLF. Therefore, Figure 15 shows the distribution of the tip leakage flow angle (TLA, defined as the angle between the leakage flow and the axial direction) at the mid-gap exit. For the prototype cascade without suction, the TLA increases first and then decreases from the LE to the TE of the blade. The axial position of the maximum TLA coincides with the position of the maximum tip load in Figure 14, and the position under large clearance is closer to the downstream and the amplitude of the maximum TLA is smaller than that under small clearance. For the case under the TCS of 0.5% h, the TLA under the suction condition is greatly reduced in the range of 4 to 62% Ca, the TLA is negative even in the range of 15 to 40% Ca, and part of TLF backflow occurs. In other ranges, the TLA is basically unchanged. The decrease of TLA indicates that the ES scheme reduces the velocity of the TLF and reduces the intensity of the TLF and TLV. For the case under the TCS of 2.0% h, the ES scheme reduces the TLA of the prototype cascade in the range of 3 to 70% Ca, and the effect on the TLA in other ranges is negligible.

From the above analysis, it can be seen that the effect of the same endwall suction scheme with the same SFR on the TLF under different TCS is all through the suction of low-energy fluid in the tip area, which reduces the blade tip load of the area covered by suction slot and its vicinity. Furthermore, the strength of the TLF and TLV, driven by the blade tip load, is weakened, and the overall performance of the cascade is improved. On the other hand, the control effect of suction on the TLF under different TCS is different. Combined with the analysis of Figure 9 and Figure 11, the SFR of 0.7% can completely absorb the TLF of the prototype cascade in the formation and development stages of the TLV under the small clearance. Thus, the onset position of the TLV moves to the downstream of the suction slot, which shortens the effective diffusion range of the TLV in the spanwise and axial direction. However, due to the larger clearance corresponding to the larger leakage flow, the same SFR only causes part of the flow near the endwall in the cascade gap be sucked away, which has no essential impact on the formation process of the TLV. Therefore, as shown in Figure 13, the suction scheme only reduces the strength of the TLV, so the improvement effect of the ES scheme designed in this paper on the overall performance of the cascade with large clearance is not as good as that with small clearance.

Finally, the effect of the ES scheme on the cascade outlet by controlling the TLF is analyzed quantitatively. Figure 16 and Figure 17, respectively, show the distribution of the pitch averaged total pressure loss coefficient and the outlet flow angle along the spanwise direction. As the suction on the upper endwall of the cascade does not affect the flow of the lower half, only the parameter distribution of the area above 50% blade span is given here. In Figure 16, the relative blade height at the beginning of loss growth under large clearance is closer to the endwall than that under small clearance, which is caused by the fact that the TLV under the large clearance has a more backward onset position and a smaller spreading range along the spanwise. The suction did not change the distribution rule of the total pressure loss coefficient along the spanwise. It mainly reduced the total pressure loss coefficient of cascade outlet in the range of 64 to 98% blade span under the TCS of 0.5% h, and 70–99% blade span under the TCS of 2.0% h. With the same SFR, both the spanwise range and the degree of loss reduction under the small clearance are larger than that under the large clearance, which corresponds to the initial analysis of the overall total pressure loss coefficient.

The outlet flow over deflects near the upper endwall under the small clearance, whereas that under deflects near the upper endwall and over deflects near 80% blade span under the large clearance. The suction also does not change the distribution rule of the outlet flow angle along the spanwise direction in Figure 17, but has different influence on the change of the outlet flow angle under the two tip clearance sizes. The outlet flow angle of the TCS of 0.5% h with suction increases above 75% blade span. The growth rate of outlet flow angle gradually increases with the blade height, and the maximum increase is obtained at the endwall. The increase of the outlet flow angle due to aspiration effectively improves the over deflection phenomenon of the outlet flow at the endwall. The outlet flow angle of the cascade with the TCS of 2.0% h after suction increases below 93% span, but decreases above 96% span. The ES scheme improves the over deflection near 80% blade span and the under deflection near the upper endwall of the outlet flow under the large clearance, which reduces the overall outlet loss and improves the aerodynamic performance of the cascade. The positive effects of the ES scheme in controlling the TLF and modifying the tip flow field are also different due to the difference of the outlet flow angle distribution between the two clearance sizes.

## 5. Conclusions

In this paper, the influence of the endwall suction scheme on the tip leakage flow and axial cascade aerodynamic performance under different tip clearance size is numerically studied. The flow field details such as tip leakage are captured and analyzed by entropy analysis and Q-criterion method. The main conclusions are as follows.

(1) With the increase of the tip clearance size, the maximum blade tip load increases, which leads to the increase of TLF and TLV intensity, and the onset position moves to the downstream. The trajectory of the TLV is far away from the SS of blade, and the blockage effect of flow passage in the tip area increases. In the range of clearance variation studied in this paper, the overall total pressure loss of cascade increases with the increase of the TCS.

(2) The endwall suction scheme adopted in this paper can effectively control the TLF by directly influencing the structure and development of the TLV, so as to modify the tip flow field and improve the aerodynamic performance of the cascade. The TCS has an important influence on the control effect of the ES scheme, and the improvement effect of suction on the performance of cascade is obviously weakened under the large clearance. Under the condition of the SFR of 0.7%, the overall total pressure loss coefficient can be reduced by 10.3%, 10.8%, and 6.0%, and 6.0%, 6.8%, and 3.2%, respectively, under the inflow incidence angle of 0° and 4°.

(3) For the small clearance, the ES scheme makes the onset position of the TLV move backward to the suction slot, reduces the effective mixing length of the TLF and main flow, and reduces the leakage loss significantly. For the large clearance, the ES scheme only weakens the intensity of the TLV, so the improvement of cascade performance is limited. The leakage flow rate and the distance from the TLV to the suction slot are the main factors that affect the control effect of the ES scheme, and also the reasons why the same SFR in current study has different effects on the performance improvement of cascades under different tip clearance sizes. In addition, the difference of the spanwise distribution of flow field parameters at outlet between the small clearance and the large clearance results in the difference of the improvement effect of suction on the flow field at different spanwise positions.

## Figures and Tables

**Figure 1 entropy-22-00128-f001:**
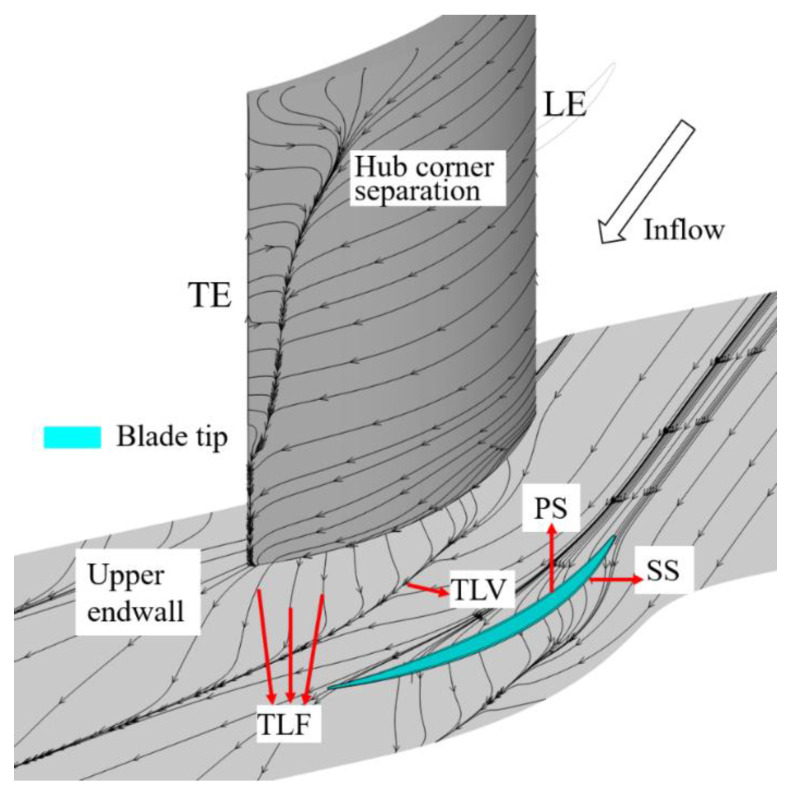
Schematic of the tip leakage flow and tip leakage vortex.

**Figure 2 entropy-22-00128-f002:**
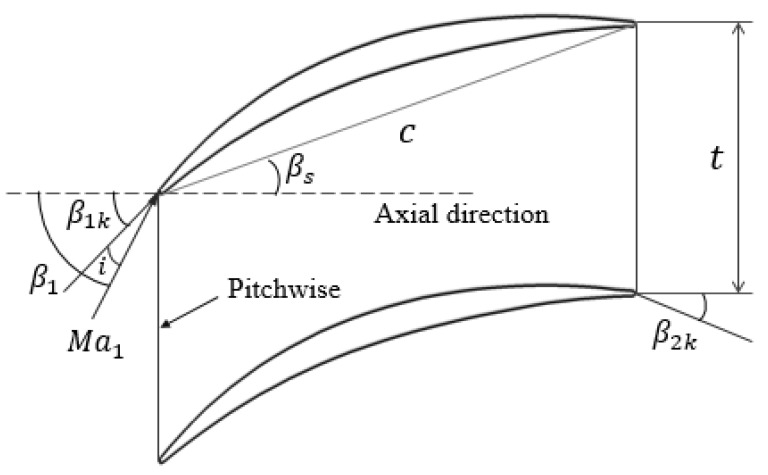
Two-dimensional cascade configuration.

**Figure 3 entropy-22-00128-f003:**
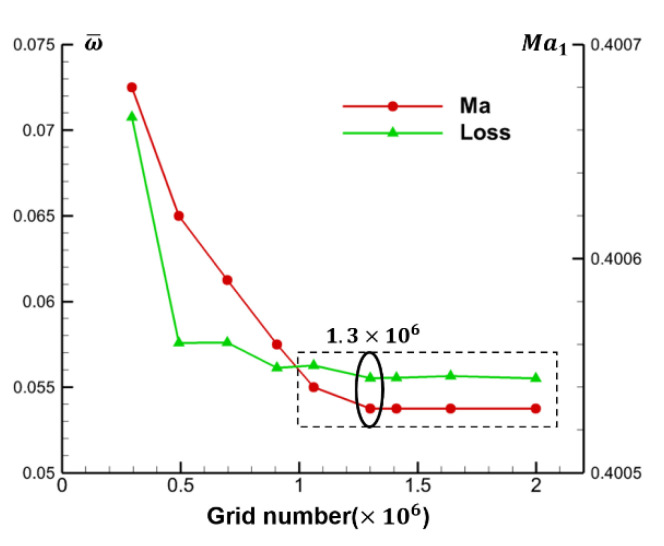
Cascade performance with different number of grid.

**Figure 4 entropy-22-00128-f004:**
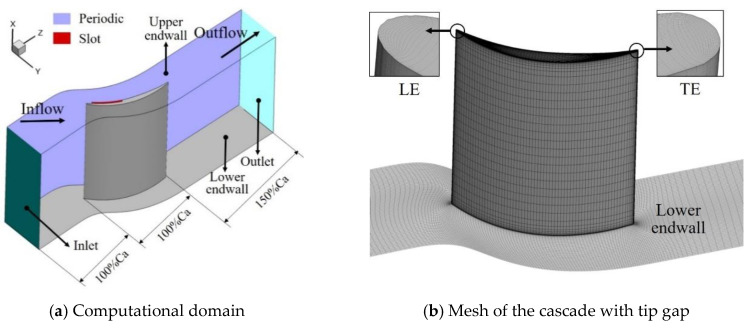
The computational domain and grids of the cascade.

**Figure 5 entropy-22-00128-f005:**
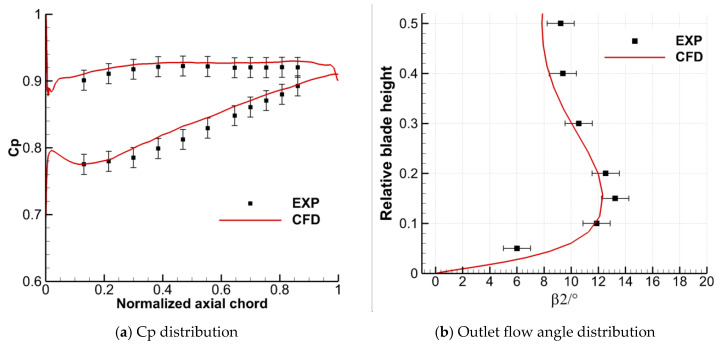
Comparison between numerical and experimental results of the original cascade without tip clearance.

**Figure 6 entropy-22-00128-f006:**
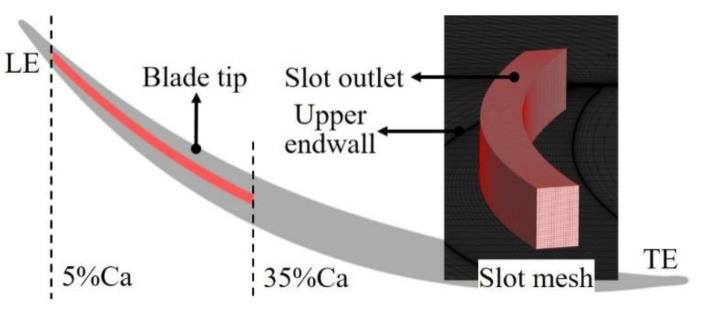
Aspiration slot arrangement and mesh of slot.

**Figure 7 entropy-22-00128-f007:**
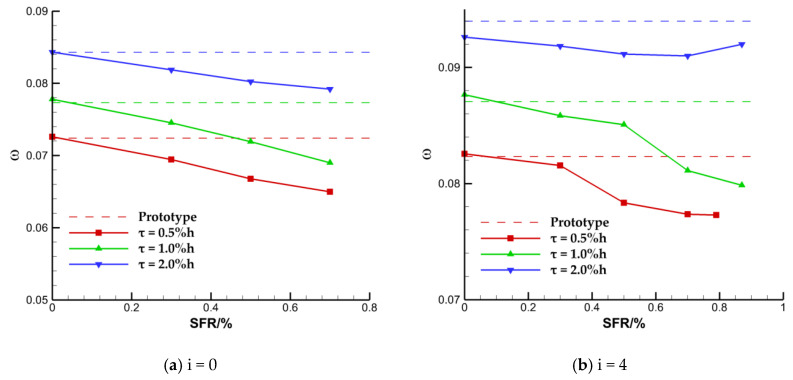
Variation of the overall total pressure loss coefficient with SFR.

**Figure 8 entropy-22-00128-f008:**
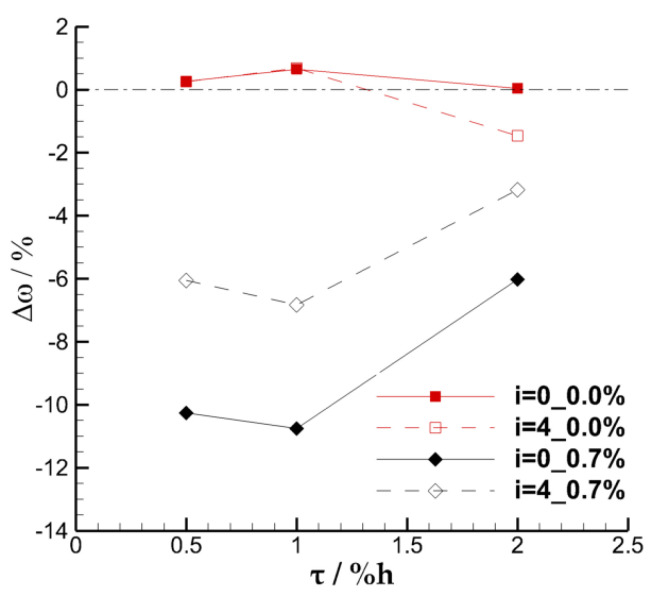
Variation of the relative change of loss coefficient with tip clearance size (TCS).

**Figure 9 entropy-22-00128-f009:**
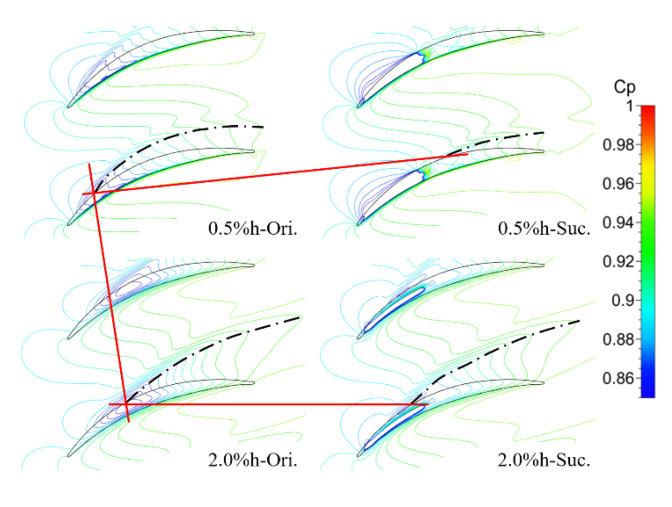
Static pressure coefficient distributions on the casing.

**Figure 10 entropy-22-00128-f010:**
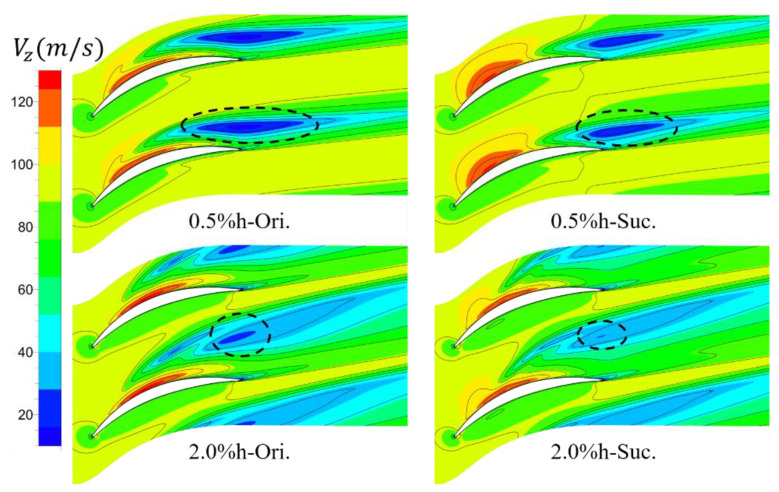
Axial velocity distributions at 97% span. The area of low speed zone caused by leakage flow is obviously reduced due to suction, and the region moves downstream under the small clearance with suction.

**Figure 11 entropy-22-00128-f011:**
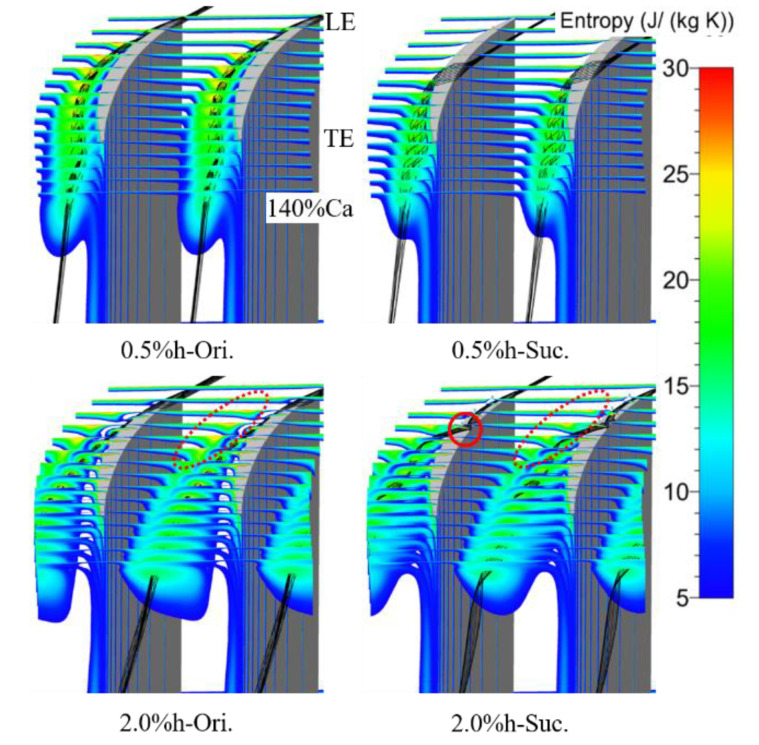
Entropy contours of tip region on axial slices and 3-d leakage streamlines. The suction makes the size of the tip leakage vortex (TLV) decrease obviously in the small clearance, and the onset position of TLV moves to the middle chord, whereas the size of TLV in the large clearance decreases slightly.

**Figure 12 entropy-22-00128-f012:**
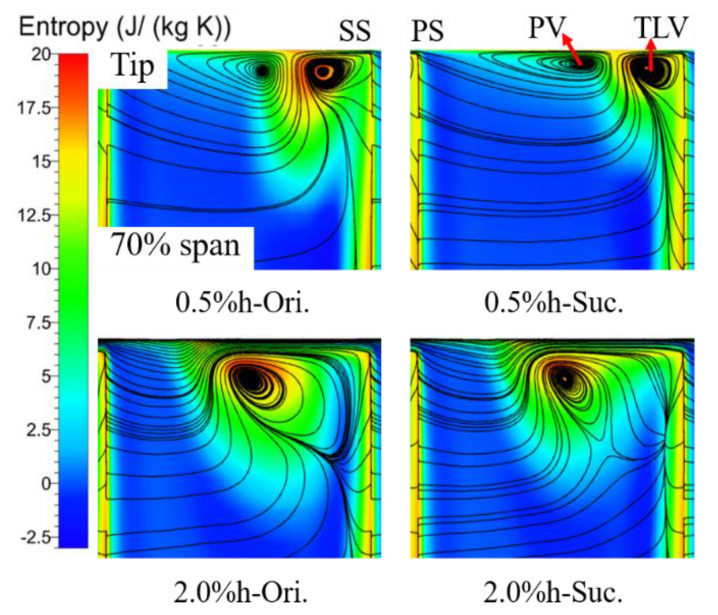
Distribution of entropy and 2-d streamlines from 70% span to tip of S3 slice at trailing edge. The PV structure can be seen in the small clearance, but it can hardly be seen in the large clearance.

**Figure 13 entropy-22-00128-f013:**
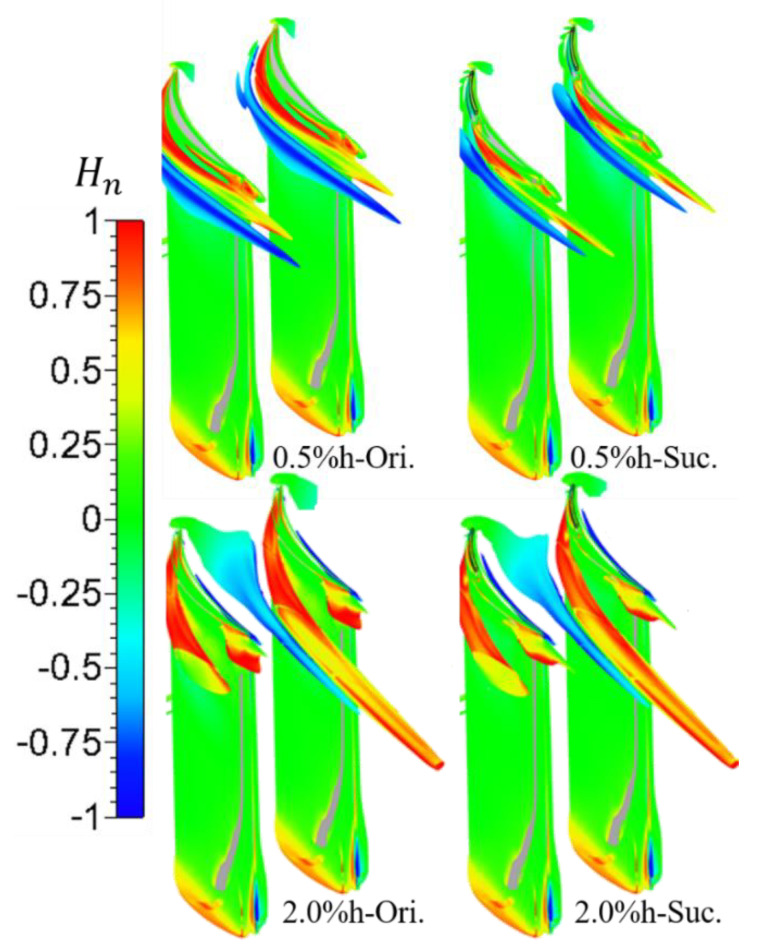
Vortices structure identified by Q-criterion. After suction, the TLV in the small clearance obviously attenuate and the onset position moves to the downstream, whereas the scale of the TLV in the large clearance has no obvious change.

**Figure 14 entropy-22-00128-f014:**
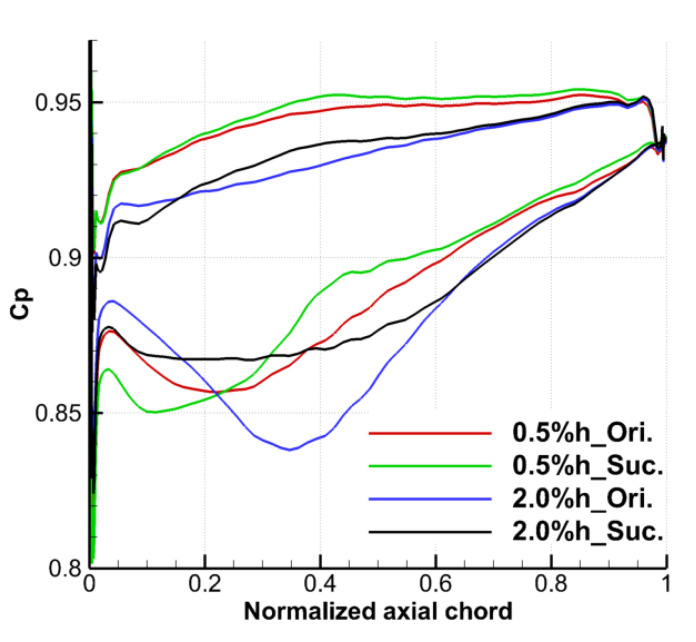
Static pressure coefficient distributions at 97% blade span.

**Figure 15 entropy-22-00128-f015:**
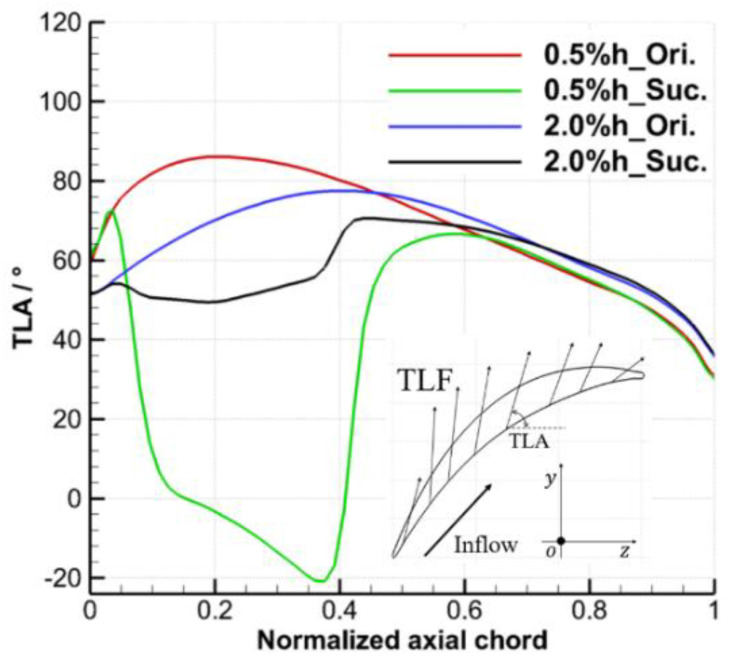
Tip leakage flow angle distributions at mid-gap exit.

**Figure 16 entropy-22-00128-f016:**
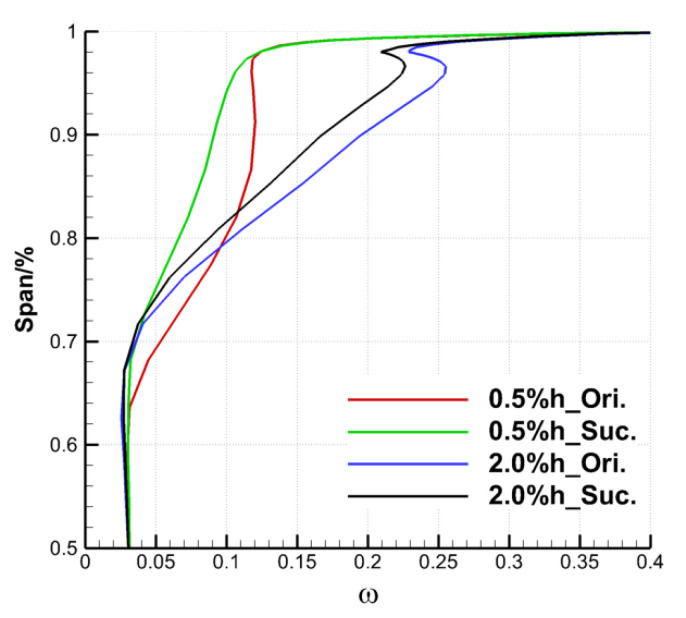
Spanwise distribution of pitch-averaged total pressure loss coefficient.

**Figure 17 entropy-22-00128-f017:**
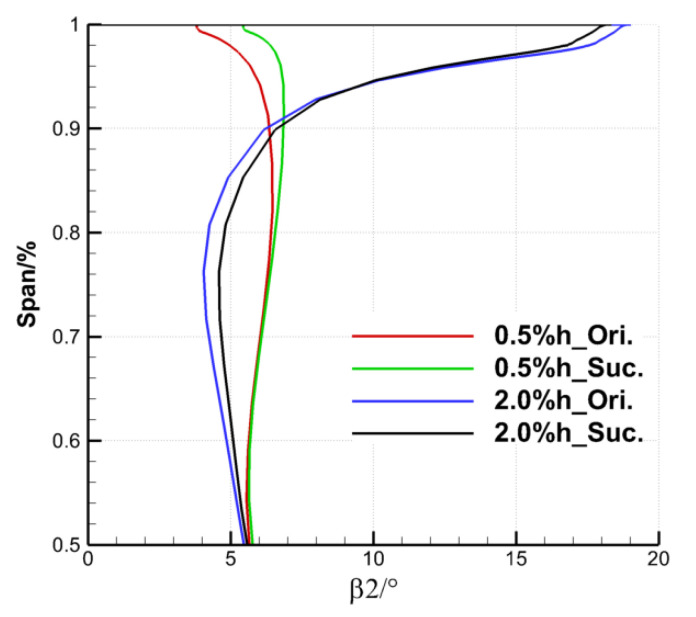
Spanwise distribution of pitch-averaged outlet flow angle.

**Table 1 entropy-22-00128-t001:** Cascade design parameters.

Parameter	Value	Unit
chord length/c	65	mm
aspect ratio	1.54	-
pitch/t	37.57	mm
solidity/τ	1.73	-
geometric inlet angle/$\beta_{1k}$	47.08	degree
geometric outlet angle/$\beta_{2k}\;$	1.98	degree
stagger angle/$\beta_{s}\;$	21.27	degree
incidence angle/i	0	degree
inlet Mach number/$Ma_{1}\;$	0.4	-

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
