# Peer review of "Control and Entropy Analysis of Tip Leakage Flow for Compressor Cascade under Different Clearance Sizes with Endwall Suction"

_entropy, 2020, doi:10.3390/e22020128_

Round 1

Reviewer 1 Report

Dear authors,

I think the Article can be published after a few revisions:

In the abstract, "h" must be defined. You should also precise somewhere that the article deals with axial-flow compressors.

The symbols in Figure 1 should be used in Table 1 (beta_s, ...). I think the "aspect ratio" is c/t ? I would call this quantity "solidity" for a blade casacade...

The first time the slot is introduced, I did not have any idea on its position and on the way it has been chosen. Could you clarify ?

The Cp that is used in Fig. 4 is not defined.

The outlet angle distribution is plotted as a function of the relative blade height in Fig. 5: why is it plotted only for the lower 50% of the blade ?

Reviewer 2 Report

The authors present a numerical study of the tip leakage flow for a compressor cascade with different clearance sizes including the effect of endwall suction. The paper is interesting and has potential to be a journal paper. However, more details are needed to improve the overall quality and the readability of the paper. In particular the results and discussion section should be improved by providing more details. The following should be addressed in the revised paper.   

(1) Line 10: Split the first sentence of the abstract in to two. So that it is clearer. Overall, the abstract could be improved by providing some more details.

(2) The introduction can be improved by providing a schematic of the flow problem addressed in the paper (end wall flow) and labelling the different features of it like the pressure surface, suction surface etc.

(3) Figure 8: It will be informative if the authors can also show the flow field in terms of the velocities and discuss it.

(4) Figure 9: This figure can be improved by making it slightly bigger so that the features in the entropy contours and the streamlines are clearly visible.

(5) Figure 11: This figure can be improved by making it bigger so that the flow structures are clearly visible and the different features are also clearer.

(6) Through out the paper the figure captions can be improved by providing a sentence or two that explains the important aspect of the result shown. For example in Figure 11, the caption can be improved by mentioning the differences in the flow structure in the different cases shown briefly in a sentence or two. This will improve the readability of the paper.

(7) There is a lot of symbols and abbreviations used in this paper. The authors should add a nomenclature that includes all the symbols and abbreviations used in the paper. It will be very useful to the readers and the reviewers.   

Round 2

Reviewer 2 Report

The authors have implemented all the corrections suggested in the review report. I can recommend publication of the paper in the journal.